# Ensembles of Convolutional Neural Networks for Survival Time Estimation of High-Grade Glioma Patients from Multimodal MRI

**DOI:** 10.3390/diagnostics12020345

**Published:** 2022-01-29

**Authors:** Kaoutar Ben Ahmed, Lawrence O. Hall, Dmitry B. Goldgof, Robert Gatenby

**Affiliations:** 1Department of Computer Science and Engineering, University of South Florida, Tampa, FL 33620, USA; lohall@usf.edu (L.O.H.); goldgof@usf.edu (D.B.G.); 2Department of Diagnostic Imaging and Interventional Radiology, H. Lee Moffitt Cancer Center and Research Institute, Tampa, FL 33612, USA; Robert.Gatenby@moffitt.org

**Keywords:** glioblastoma multiforme, brain tumor, survival prediction, magnetic resonance images, BraTS, artificial intelligence, machine learning, deep learning

## Abstract

Glioma is the most common type of primary malignant brain tumor. Accurate survival time prediction for glioma patients may positively impact treatment planning. In this paper, we develop an automatic survival time prediction tool for glioblastoma patients along with an effective solution to the limited availability of annotated medical imaging datasets. Ensembles of snapshots of three dimensional (3D) deep convolutional neural networks (CNN) are applied to Magnetic Resonance Image (MRI) data to predict survival time of high-grade glioma patients. Additionally, multi-sequence MRI images were used to enhance survival prediction performance. A novel way to leverage the potential of ensembles to overcome the limitation of labeled medical image availability is shown. This new classification method separates glioblastoma patients into long- and short-term survivors. The BraTS (Brain Tumor Image Segmentation) 2019 training dataset was used in this work. Each patient case consisted of three MRI sequences (T1CE, T2, and FLAIR). Our training set contained 163 cases while the test set included 46 cases. The best known prediction accuracy of 74% for this type of problem was achieved on the unseen test set.

## 1. Introduction

A glioma is the most common primary tumor of the brain. It originates from glial cells. About 80% of malignant brain tumors are gliomas. According to the 2021 WHO Classification of Tumors of the Central Nervous System [1], glioblastoma is a sub-category of adult-type diffuse gliomas. It is considered a WHO grade IV glioma which represents approximately 57% of all gliomas and 48% of all primary malignant central nervous system (CNS) tumors [2]. Glioblastoma is highly invasive and can quickly spread throughout every region of the brain. Standard therapy consists of resection (surgery) followed by a combination of radiation and chemotherapy. Despite aggressive multimodal treatment, patients diagnosed with glioblastoma multiforme (GBM) have a dismal prognosis with an average survival time of slightly more than 1 year (15–16 months) [3].

Neuro imaging remains the cornerstone of tumor diagnosis and assessment during therapy. While treatment response continues to simply be assessed on changes in tumor size, new image analytic tools have been introduced to access additional information from clinical scans. Radiomics is a non-invasive method for the extraction of quantitative features from medical imaging that may not be apparent through traditional visual inspection [4]. By far, the most common neuro-imaging tool is magnetic resonance imaging (MRI). MRI provides an isotropic three dimensional (3D) picture of the brain with excellent soft tissue contrast and resolution without potentially harmful radiation. Typically, MRI imaging is performed in three different planes; axial, coronal, and sagittal. Figure 1 shows a slice of the human brain in three planes. Commonly used MRI sequences include contrast enhanced T1-weighted (T1CE), T2-weighted, and fluid attenuation inversion recovery (FLAIR) as shown in Figure 2. These sequences, because they are sensitive to different components of the tumor biology, play an integral role in providing refined anatomic and physiological detail. For instance, T1-weighted post-contrast images are sensitive to the enhancing core and necrotic tumor core, whereas FLAIR and T2 highlight the peritumoral edema.

Recently, deep learning methods have been applied in computer vision and biomedical image analysis to improve the ability to extract features from images automatically and increase predictive power. They rely on advances in neural network architectures, GPU computing, and the advent of open source frameworks such as Tensorflow [5] and Pytorch [6]. When trained with sufficient data, deep neural networks can attain high accuracy in medical image analysis. The present state of the art is largely dominated by convolutional neural networks (CNN) with a number of effective architectures such as VGG-Net [7], Inception networks [8], and ResNet [9]. Instead of manually engineering features, convolutional neural networks allow automatic convolutions to extract an enormous number of features from input images. Recent review works [10] show the domination of convolutional neural network techniques applied to brain magnetic resonance imaging (MRI) analysis compared to other approaches.

Survival time after a patient is diagnosed with high grade glioma depends on several factors such as age, treatment, tumor size, behavior, and location, as well as histologic and genetic markers [11]. Integration of automated image analytics can provide insights for diagnosis, therapy planning, monitoring, and prognosis prediction of gliomas. A new study [12] by UT Southwestern shows deep learning models can automatically classify IDH mutation status in brain gliomas from 3D MR imaging with more than 97% accuracy. This technology can potentially eliminate the current need for brain cancer patients to undergo invasive surgical procedures to help doctors determine the best treatment for their tumors. This represents an important milestone towards non-surgical strategies for histological determination especially in highly eloquent brain areas such as the brain stem.

Applying deep learning algorithms to multi-sequence MR images is challenging due to several factors. First, the size of the 3D data. Second, the difficulty in designing an appropriate neural network architecture for the desired objective. Third (and most critically), the limited clinical knowledge of deep learning experts and the limited availability of large medical image datasets with labels. Finally, predictive power of deep convolutional neural network models is significantly enhanced by a large training dataset. Using a small number of cases to train a model using deep learning can result in over-fitting the training data such that generalization to future cases is poor.

Here, we address the issues of limited and heterogeneous data sets in medical imaging using an ensemble learning method applied to survival time prediction from glioblastoma patient data. Its efficacy is demonstrated through application to MR images from the BraTS dataset.

We present our investigation as follows: Section 2 discusses related research studies and recent trends of survival analysis and radiomics using deep learning. In Section 3, we describe our proposed survival prediction system in detail including data acquisition, preparation and augmentation, region of interest segmentation and model construction, and training. Our experimental results are presented in Section 5. In Section 6, the results are discussed and conclusions are drawn from experimental outcomes.

## 2. Related Work

The task of survival prediction of glioma from MRI images is challenging. Several studies applying various methods and approaches using the BraTS dataset are reviewed in this section.

Authors of [13,14,15] proposed the use of some handcrafted and radiomics features extracted from automatically segmented volumes with region labels to train a random forest regression model to predict the survival time of GBM patients in days. These studies achieved respectively 52%, 51.7%, and 27.25% accuracy on the validation set of the BraTS (Brain Tumor Image Segmentation) 2019 challenge.

Authors of [16] performed a two category (short- and long-term) survival classification task using a linear discriminant classifier that was trained with deep features extracted from a pre-trained convolutional neural network (CNN). The study achieved 68.8% accuracy when doing 5-fold cross validation on the BraTS 2017 dataset.

Ensemble learning was used by authors in [17]. They extracted handcrafted and radiomics features from automatically segmented MRI images of high grade gliomas and created an ensemble of multiple classifiers, including random forests, support vector machines, and multilayer perceptrons, to predict overall survival (OS) time on the BraTS 2018 testing set. They obtained an accuracy of 52%.

The authors of [18] achieved first place in the BraTS 2020 challenge for the overall survival prediction task (61.7% accuracy). They extracted segmentation features along with patient’s age to classify the patients into three groups (long-, short- and, mid-term survivors) using an ensemble of a linear regression model and random forests classifier.

Over the last decade, there was increasing interest in ensemble learning for tumor segmentation tasks as well. Ensemble learning was ubiquitous in the BraTS 2017–2020 challenges, being used in almost all of the top-ranked methods. The winner of the BraTS 2017 challenge for GBM tumor segmentation [19] was an ensemble of two fully convolutional network models (FCN), and a U-net each generating separate class confidence maps. Then, each class was created by averaging the confidence maps of the individual ensemble models for each voxel. This study reached dice scores of 0.90, 0.82, and 0.75 for the whole tumor, tumor core, and enhancing tumor, respectively for the BraTS 2017 validation set. Authors in [20] built an ensemble of UNet-based deep networks trained in a multi-fold setting to perform segmentation of brain tumors from the T2 Fluid Attenuated Inversion Recovery (T2-FLAIR) sequences. They achieved a dice score of 0.882 for the BraTS 2018 set.

## 3. Materials and Methods

### 3.1. Data Acquisition and Preparation

#### 3.1.1. BraTS Dataset

The Brain Tumor Segmentation (BraTS) Challenge is critical for benchmarking and has largely contributed to advancing machine learning applications in glioma image analysis. The BraTS challenge was first held in 2012 and has taken place annually as part of the Medical Image Computing and Computer Assisted Intervention (MICCAI) conference ever since. It focuses on evaluating state-of-the-art methods for the segmentation of brain tumors in multimodal magnetic resonance imaging (MRI) scans. The glioma sub-regions considered for segmentation evaluation are: (1) The enhancing tumor (ET), (2) the tumor core (TC), and (3) the whole tumor (WT). The enhancing tumor sub-region is described by areas that are typically hypo-intense in T1CE when compared to T1. The tumor core (TC) consists of the bulk of the tumor, which is what is typically resected. The TC entails the ET, as well as the necrotic (fluid-filled) and the non-enhancing (solid) parts of the tumor. The WT corresponds to the complete extent of the disease, as it consists of the TC and the peritumoral edema (ED) [21].

To evaluate proposed segmentation methods, the participants are asked to upload their segmentation labels for regions in the image (e.g., edema, tumor) as a single multi-label file in the NIfTI format. Then metrics like dice score and the Hausdorff distance are reported to rank teams performances. In 2017, the BraTS challenge started to include another task, the prediction of overall survival time. Participants needed to use their produced segmentation in combination with other features to attempt to predict patient overall survival. The evaluation assessment of this task was based on the accuracy metric of a three category classification (long-survivors (e.g., >15 months), short-survivors (e.g., <10 months), and mid-survivors (e.g., between 10 and 15 months) ).

The Brain Tumor Segmentation (BraTS) challenge dataset is the largest publicly available glioma imaging dataset. It includes multi-institutional pre-operative clinically-acquired multi-sequence MRI scans of glioblastoma (GBM/HGG) and lower grade glioma (LGG) with overall survival data. Images were acquired from different institutions using MR scanners with different field strengths (1.5 T and 3 T). Segmentation ground truth labels are done by expert board-certified neuroradiologists. Scans are provided in NIfTI format and have (a) native (T1) and (b) post-contrast T1-weighted (T1CE), (c) T2-weighted (T2), and (d) T2 Fluid Attenuated Inversion Recovery (T2-FLAIR) sequences with 1–6-mm slice thickness. The BraTS data was made available after its pre-processing, i.e., co-registered to the same anatomical template, interpolated to the same resolution, and skull-stripped. The overall survival data is defined in days.

#### 3.1.2. Data Acquisition

In this study we used subsets of glioblastoma (GBM) cases from the BraTS 2019 training dataset. We randomly split the data into train/test subsets. The training subset consisting of 163 cases and the testing set had 46 cases. Among the 163 cases, there were 81 long-term and 82 short-term survival cases based on a 12 month cut-off (which is the middle threshold of the mid-term survivors class in BraTS data). Among the 46 test cases, there were 23 long-term and 23 short-term survivors. The data can be downloaded online from the CBICA’s Image Processing Portal (IPP) after requesting permission from the BraTS team. Figure 3 shows the Kaplan–Meier survival graph corresponding to the training dataset.

#### 3.1.3. Data Pre-Processing and Augmentation

The first step is tumor segmentation, we decided to use the publicly available pre-trained model of Wang et al. [22] to automatically segment the training and testing set used in this study. The segmented volumes were used as a starting point for our task of survival class prediction.

Each MRI volume is high resolution with a size of 240×240×155 which makes it difficult to fit into the GPU memory. To solve this issue, we had to remove empty voxels and null slices that are unnecessary for building a good predictive model.

Data augmentation is a fundamental way to help reduce model over-fitting when dealing with a small amount of training data. Hence, we performed transformations to each 2D slice consisting of vertical flipping and rotations of 10 degrees incrementally from 0 to 180 degrees. In addition, we applied elastic deformation by generating smooth deformations using random displacement fields that are sampled from a Gaussian distribution with a standard deviation of 17 (in pixels). The new values for each pixel were then calculated using bicubic interpolation. The augmentation was applied to each 2D slice which then gets stacked back in to form a 3D cube. The transformations resulted in increasing the size of training data from 163 images per sequence to roughly 10,000 augmented images, including the original images. Then all the images were resized to a uniform size of 160×160×110. Furthermore, we used the Keras [23] data generator to feed data by batches to our model instead of loading all data into memory at once. In addition, before feeding input images to the model, as a part of preprocessing, the images were normalized to have zero mean and unit standard deviation.

### 3.2. Overall Survival Prediction System

Our survival prediction system is next described in detail including data acquisition, preparation and augmentation, region of interest segmentation, model construction, and training.

#### 3.2.1. System Outline

As shown in Figure 4, the overall work flow of our system consists of three stages. Firstly, the starting point was the raw 3D MRI images acquired from the BraTS platform after securing permission. The images were sent as inputs to the pre-trained segmentation network [22] resulting in a 3D segmented tumor in the NIfTI format. Image augmentation and pre-processing were done before the inputs were sent to our 3D convolutional neural network for automatic feature extraction and classifier training. Finally, the end point is the overall survival prediction. Ensemble learning was used to evaluate the prediction performance of the trained networks. More details on each phase are provided in the following sub-sections.

#### 3.2.2. Snapshot Learning for Survival Prediction

An ensemble of deep neural networks is known to be more accurate and robust. However, training multiple deep networks is computationally expensive. Huang et al. [24] introduced snapshot learning which is a fast and effective way of creating an ensemble of models without additional training costs by saving one model’s weights snapshots at different points over the course of training. Our preliminary results in [25] showed the effectiveness of an ensemble of snapshots in boosting the prediction performance using our limited size dataset. Snapshots of the model are saved at every training iteration then a selected subset of them are used to evaluate the model’s prediction performance on a separate testing set. Majority voting was used to combine their predictions. The snapshots of the model are each evaluated and a classification outcome is obtained for each of the 46 test cases. Each sample can be correctly or incorrectly classified by a particular snapshot model. The predictions each provide a vote and our combination approach is to take the class with the most votes for each test example. The voting rule is as follows: If more than 50% of the snapshot models have correctly classified test example X then it is correctly classified. Otherwise, if ≤50% of the votes correctly classified the example then it is considered incorrectly classified. The overall final prediction accuracy will be the percentage of correctly predicted test examples (46).

### 3.3. Main CNN Model Construction and Training

Finding the most efficient network parameters for a given dataset is important. It is known that parameters like batch size, number of epochs, regularization, and learning rate can significantly affect the final performance. There is no predefined method to select the ideal architecture and parameters. Therefore, we attempted multiple combinations of settings and parameters. Our model consists of three 3×3×3 convolutional layers (Conv) with a stride of 1×1×1 size and ReLU activation function followed by a 2×2×2 max-pooling layer with a stride of 1×2×2. The first two convolutional layers have 20 filters and the third has 40. Padding was used for the convolutional layers. A fully connected layer (FC) of 64 units follows a global average pooling operation. Finally, an output with sigmoid activation is used to generate a prediction. To reduce overfitting, a dropout of 50% was added after the fully connected layer. The developed model consists of a total of 35,709 parameters.

The implementation of the model was carried out with Keras 2.3.1 using TensorFlow 1.2 as backend. Code is available on GitHub at https://github.com/kbenahmed89/Glioma-Survival-Prediction, accessed on 14 December 2021.

## 4. Experiments and Results

The experimental settings and results of this work are reported in this section. Measures include accuracy, area under the ROC curve (AUC), sensitivity, and specificity.

### 4.1. Parameter Exploration Results

In this section, we report some preliminary experiments performed to determine the best parameters for our dataset. Here, for simplicity, we only used a T1 contrast enhanced (T1CE) sequence as input.

#### 4.1.1. Choice of Snapshots

The original snapshot learning method showed using the cosine annealing cyclic learning rate was most effective. It works by saving snapshots at the end of each cycle (X number of training epochs). Here, we compare the performance results of the following: The original method versus using fixed learning rate and selecting snapshots based on the training accuracy evolution. When not using cyclic learning the stochastic gradient descent optimizer was used for training the models with the starting learning rate set to $10^{-3}$ and momentum set to 0.9. The network state was saved at each epoch. We selected snapshots based on the training accuracy graph. The accuracy of the model starts to increase rapidly in the beginning due to the high initial learning rate. Then, it slows down and keeps increasing at a lower rate and with high stability. During the last period of training, the training accuracy has already reached 100% thus over-fitting the training data. Therefore, we decided to avoid the last set of training epochs and only focus on the earlier phase of training. To create an ensemble of snapshots, we picked the weights of every five epochs to allow a fair amount of change in the network weights. We go back 5 steps from when the training accuracy reaches 80% (epoch 30) and then proceed backwards to choose every fifth epoch as a snapshot (epochs 30, 25, 20, 15, and 10). The performance results of testing the individual five snapshots on the 46 example testing dataset are shown in Table 1 along with the results of the majority voting-based ensemble of snapshots.

To see if the original approach to snapshots might be better, we trained using a cosine annealing cyclic learning rate with a maximum learning rate of $10^{-3}$. Training was done for 500 epochs and snapshots saved at the end of each of the 10 cycles, which were 50 epochs long. The majority voting-based ensemble of the last five snapshots saved at the end of the last five cycles achieved a 52.17% accuracy on the unseen test set of 46 test cases using only T1CE MRI modality. This was not as good as our approach of choosing snapshots from the training data. Of course, it is possible that a different set of parameters would result in better performance, but we had little data to test with and wanted to leave the test set alone until parameters were chosen.

Based on this experiment, we decided that our choice of snapshots based on the training graph works best for this particular dataset. Therefore, we used this snapshot choice method for the remainder of the paper.

#### 4.1.2. Ensemble of Convolutional Neural Networks vs. Snapshot Learning

This experiment consisted of the performance comparison of an ensemble of five CNNs versus an ensemble of five snapshots from the same CNN. The CNNs have the same main architecture described in Section 3.3 but different random initializations. We decided to test at the epoch where the models reach a training accuracy of 80% to avoid overfitting. The results are presented in Table 2. The results are roughly equivalent to the ensemble of 5 fully trained models with much less training time.

#### 4.1.3. Comparison of 3D T1CE vs. 2D T1CE MRIs

In this experiment, due to high computational costs and time required for 3D image processing, we investigated whether the third dimension added significant value over the 2D analysis of the images. We decomposed each 3D scan into 155 separate slices. Then, we manually looked at each slice and chose the one with the largest visible tumor area. Consequently, the new training set in this experiment consists of 163 images of size 240×240×1. In order to increase the size of the dataset, image augmentation was also done using the same transformations explained in Section 3.1.3. We created a 2D version of the main CNN and kept the same architecture and parameters. For this dataset, after observing the training accuracy curve, we noticed that the model reaches 80% training accuracy quickly (epoch 28). Therefore, we decided to go back 5 steps from when the training accuracy reaches 90% (epoch 59) and then proceed backwards to choose every fifth epoch as a snapshot. Thus, epochs 59, 54, 49, 44, and 39 were used. The performance results of snapshots ensemble using the 2D CNN model is compared to the 3D CNN in Figure 5.

### 4.2. Combination of Multi-Sequence Data for Survival Prediction

In this section we describe two methods attempted in order to merge the three MRI sequences (T1CE, T2, and FLAIR) to create a multi-sequence dataset for our model.

#### 4.2.1. Ensemble of Ensembles Using T1CE, T2, and FLAIR MRIs

Before merging the three sequences, we first trained our CNN model that has the same settings and using the same training procedure described in Section 3.3, this time with the other two sequences (T2 and FLAIR) as inputs instead of T1CE. Results are reported for each sequence individually.

To create a combined decision, an ensemble of ensembles approach in which 5 snapshots from each of the three sequences T1CE, T2, and FLAIR together participate to build a final outcome prediction was used. To create each of the 5 initial ensembles, we used the 5 snapshots of each sequence that were trained individually. In a given ensemble, for each sample, one snapshot from each of the three sequences gets to vote. We repeat the process for all the 5 snapshots. We then combine the 5 voted decisions and again use majority voting at a sample level to form the final outcome.

#### 4.2.2. Multi-Branch CNN Training with Multi-Sequence 3D MRIs

In this experiment, we combine the three sequences (T1CE, T2, and FLAIR) together to train a multi-branch 3D CNN and test it on the multi-sequence MRI test set. The architecture of the CNN is illustrated in Figure 6.

Separate CNN models operate on each sequence where each CNN has two 3D convolutional layers each followed by a max-pooling layer. Then, the results from the three models are concatenated for interpretation and ultimate prediction. The CNN was evaluated using a snapshot ensemble with max voting as explained in Section 5. Here, we used a lower learning rate ($10^{-4}$) than in the previous experiments. Thus, the model makes small changes between epochs. Therefore, we decided to enlarge the spacing between chosen epochs. We go back 5 steps from when the training accuracy reaches 90% (epoch 183) and then proceed backwards to choose every 25th epoch as a snapshot (epochs 183, 158, 133, 108, and 83). The performance results of the multi-branch CNN are presented in Figure 7 and compared to the performance outcome of each individual sequence and to the first ensemble method described in Section 4.2.1.

### 4.3. Comparison with Other Methods

To further assess the prediction performance of our method, we compared our approach to the method proposed in [16]. In their paper, the authors used the BraTS 2017 training dataset with a threshold time of 18 months to classify cases into short-term and long-term survival. They extracted volumetric and location features from the 163 input images and trained a logistic regression classifier. They achieved an accuracy of 69% using 5-fold cross validation. The BraTS 2017 training dataset is a subset of the 2019 training dataset used here. To compare our results with their method, we used the same 163 cases and we also used an 18-month threshold to divide the data into two classes, short- and long-term survival. We then augmented the data and partitioned it to do the five-fold cross validation experiment. Our snapshot ensemble was used on each of the five folds. The average accuracy of the five folds was 6% higher than in [16] at 75%.

## 5. Discussion

As demonstrated in Table 2, the overall survival prediction performance of the ensemble of CNNs method outperformed individual CNNs. It was 67% accurate whereas the maximum we could get if we only use one individual CNN was 63%.

From Table 1, a snapshot ensemble indeed can perform as well as an ensemble of multiple CNNs and is faster and less expensive to train. Clearly, training five separate models (sequentially) is five times slower than saving snapshots of a single model. Using a snapshots ensemble, we achieved a prediction accuracy of 70%. We can also see that the performance of an ensemble is sometimes the same as only one snapshot, but it usually outperforms individual snapshots. Generally, if it can not improve the performance, it does not cause a clear decrease in accuracy.

As seen in Figure 5, comparing these results to the performance of a CNN trained using 2D images, we can conclude that even though 3D processing is expensive and time consuming, it brings relevant extra information to increase the model’s performance from 61% to 70%.

Figure 7 summarizes the performance results of models trained using T1CE, T2, and FLAIR sequences standalone along with the results of the two attempted combining methods. The accuracy of the voting ensemble for T2 was 63% and for FLAIR was 65%. The 70% prediction accuracy using the T1CE sequence was slightly higher. This result probably reflects the sensitivity of T1CE to the size of the tumor core area, which is probably the most important biological predictor of overall survival in GBM patients. Experiments with combining the three sequences together were done. The first method of combination we explored was voting an ensemble of sequences. This ensemble achieved a 72% prediction accuracy. The second combining method involved creating a multi-branch CNN where each sequence was sent as input to a branch then the results were combined in the final layer. The prediction accuracy of the ensemble using this multi-branch CNN reached 74%.

The result of the two experiments with sequence combination demonstrates the potential power in the combination of multiple sequences where each contributes unique information relevant to the overall survival prediction. We can also conclude that all the tumor areas are important to determine the survival of the patient. Though perhaps the tumor core area contains the most information.

Since the BraTS testing dataset is not released by the BraTS challenge team and the validation set is only available at the time of the challenge, we were unable to directly compare our method to those applied to BraTS test/validation set. However, our prediction accuracy of 74% on an unseen testing set from BraTS 2019 suggests good generalization ability to other datasets. In addition, in the experiment conducted in Section 4.3, our method achieved a prediction accuracy of 75%, thus outperforming the approach proposed in [16], which had 69% accuracy on the same dataset.

## 6. Conclusions and Future Work

In summary, we demonstrated an overall survival prediction for glioblastoma patients at 74% accuracy using a multi-branch 3D convolutional neural network, where each branch was a different MR image sequence. The system classified patients into two classes (long- and short-term survival). In a comparison on the same data with the best known predictor, our approach was 6% more accurate. An important contribution of this work is the effective use of snapshot ensembles in a novel fashion towards a solution to the limited availability of labeled images in many medical imaging datasets. Snapshots are a fast, low cost way to generate an ensemble of classifiers.

Based on recent studies [26,27,28,29,30,31,32,33,34], improved communication and information exchange between radiology and histopathology is needed more than ever. Many papers have shown benefits in integrating several markers for more accurate OS prediction of patients with glioma. A study [29] used deep artificial neural networks on histopathologic images of gliomas and found an association of OS with several histopathologic features, such as pseudopalisading and geographic necrosis and inflammation. Recent work [26,32,33] has shown the integrated potential of MR and histopathologic images, which has provided diagnostically relevant information for the prediction of OS in glioma. Similarly, it has been demonstrated that combining radiomics with multiomics can offer additional prognostic value [34]. Another study confirmed higher prediction accuracy when a combination of histopathologic images and genomic markers (isocitrate dehydrogenase [IDH], 1p/19q) was used [27]. Moreover, it has been shown that the addition of a radiomics model to clinical and genetic profiles improved survival prediction on glioblastoma when compared with models containing clinical and genetic profiles alone [28].

In future, we intend to create a more advanced version of our deep learning model combining radiologic-based features from MR imaging with pathologic, clinical, and genetic markers to develop more informed and better predictors of overall survival of GBM patients.

This study has the limitation of using a pre-operative dataset to perform OS prediction. This can result in leaving out several prognostic factors that are available after histological confirmation. In research, the most common prognostic factors for survival in glioblastoma patients were found to be: The age of patients, extent of resection, recursive partitioning analysis (RPA) class [35], performance status (using Karnofsky Performance Scale (KPS) or the Eastern Cooperative Oncology Group (ECOG)/World Health Organization (WHO) performance status) [36], and postoperative chemotherapy and/or radiotherapy [37].

In addition to IDH mutations [38] and 1p/19q codeletion [39], the methylation status of the O6-methyl guanine DNA methyltransferase (MGMT) gene promoter has been shown to be a strong predictor of the survival of glioblastoma patients [40]. There exists other prognostic molecular markers for which diagnostic evaluation is not routinely performed such as G-CIMP methylation, TERT promoter mutations, EGFR alterations, BRAF V600E mutations, Histone mutations, and H3K27 mutation, which can occur in histone H3.1 or H3.3 [41].

On the other hand, it may be possible to identify additional prognostic information using image analysis of the post-operative or post-treatment MRIs. A recent study has demonstrated an association between post-operative residual contrast-enhancing tumor volume in the post-surgical MRI and overall survival in newly diagnosed glioblastoma [42]. Furthermore, in our recent work [43], we showed that deep features extracted from post-treatment MRI scans, which were obtained during or after therapy, can entail relevant information for overall survival prediction. Therefore, to lend further validity to our model, we aim to combine data before and after therapy or resection, if available.

Recently, there has been increased interest in integrating non-invasive imaging techniques with clinical care in brain tumor patients. For instance, the use of amino acid PET has been shown to better identify the most biologically aggressive components of heterogeneous low and high-grade glioma [44,45]. As another future work direction, we may explore the potential of non-invasive metabolic imaging as complementary to conventional MRI to predict survival outcomes of GBM patients using convolutional neural networks.

## Figures and Tables

**Figure 1 diagnostics-12-00345-f001:**
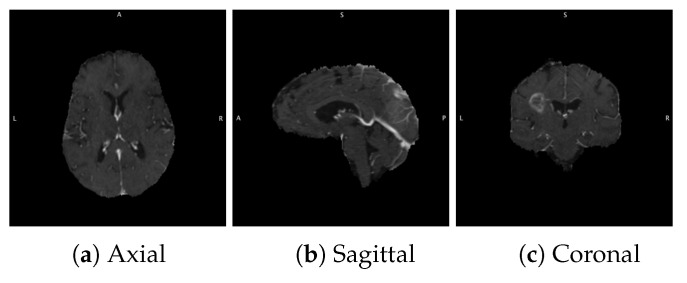
Different planes of brain MRI with the T1CE sequence.

**Figure 2 diagnostics-12-00345-f002:**
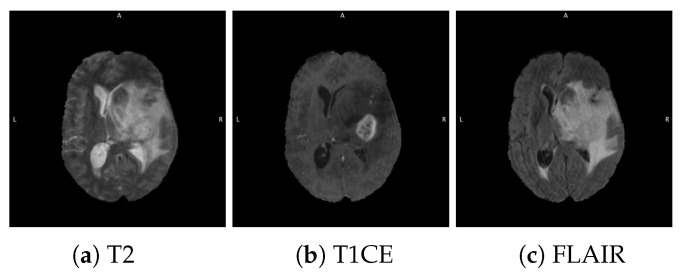
Different sequences of brain MRI in the axial plane.

**Figure 3 diagnostics-12-00345-f003:**
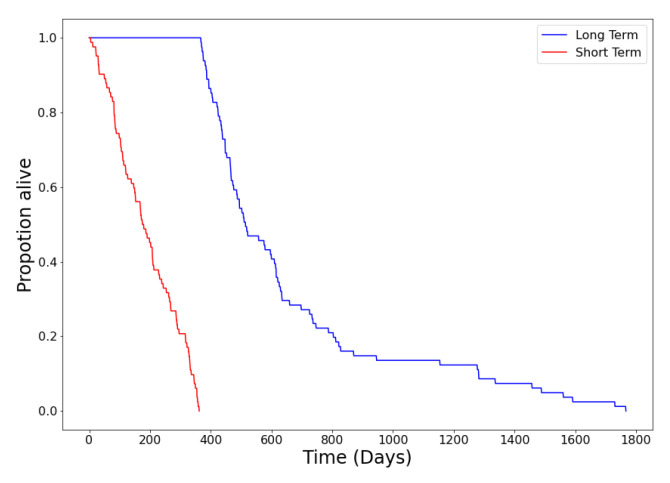
Kaplan–Meier survival graph of the training set.

**Figure 4 diagnostics-12-00345-f004:**
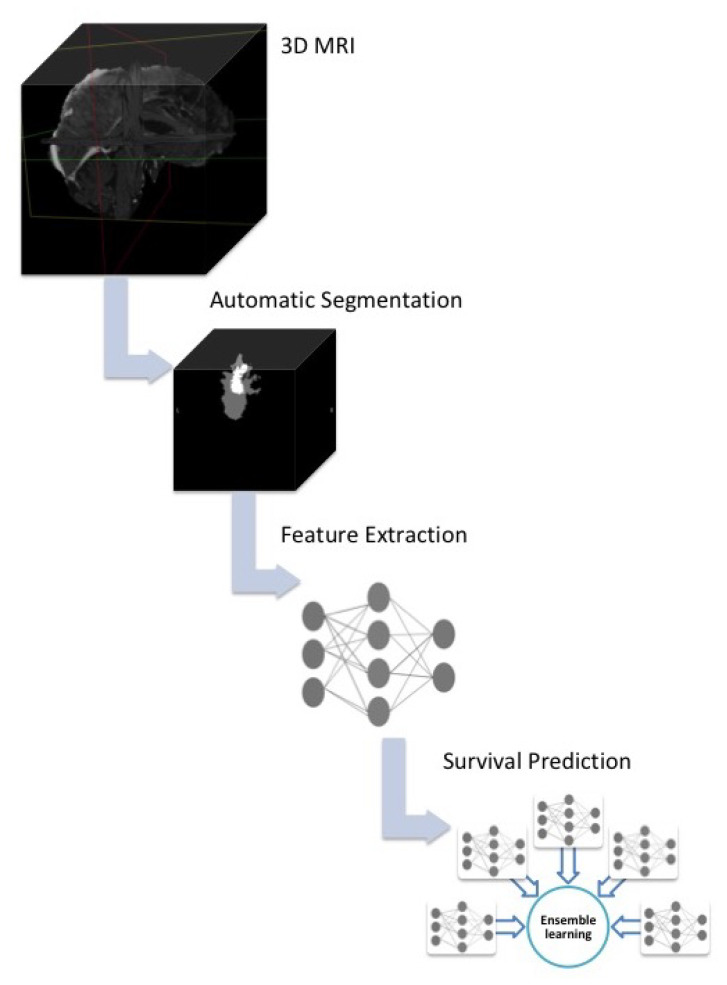
Pipeline of proposed overall survival prediction system.

**Figure 5 diagnostics-12-00345-f005:**
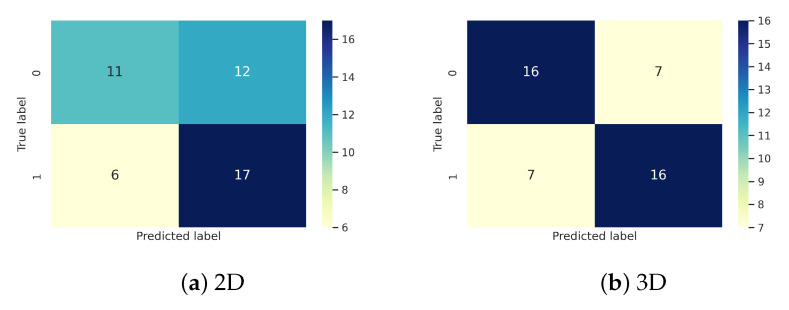
Confusion matrix comparison of 3D vs 2D CNN using the T1CE sequence.

**Figure 6 diagnostics-12-00345-f006:**
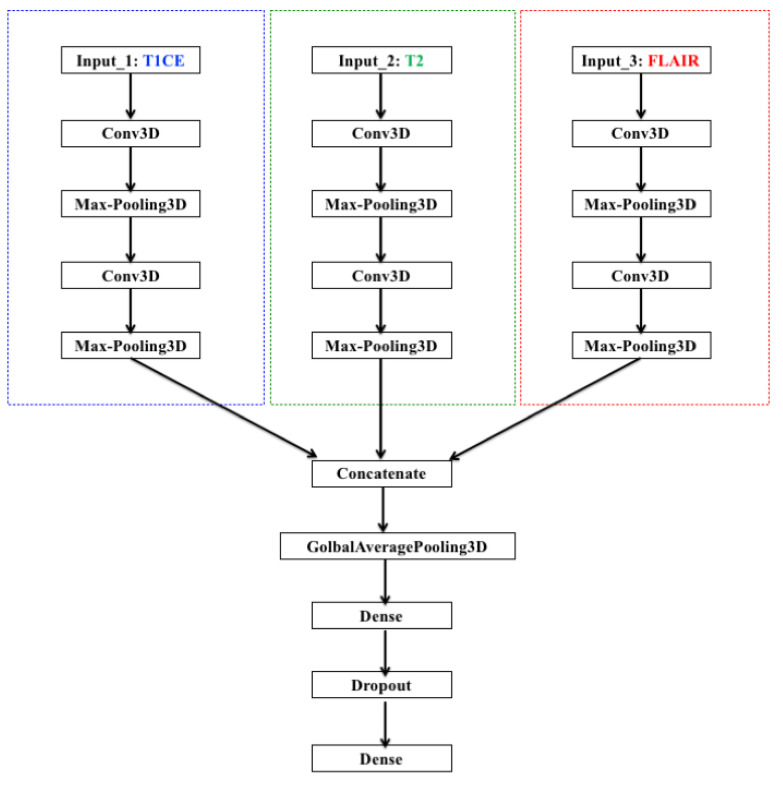
Multi-branch CNN architecture.

**Figure 7 diagnostics-12-00345-f007:**
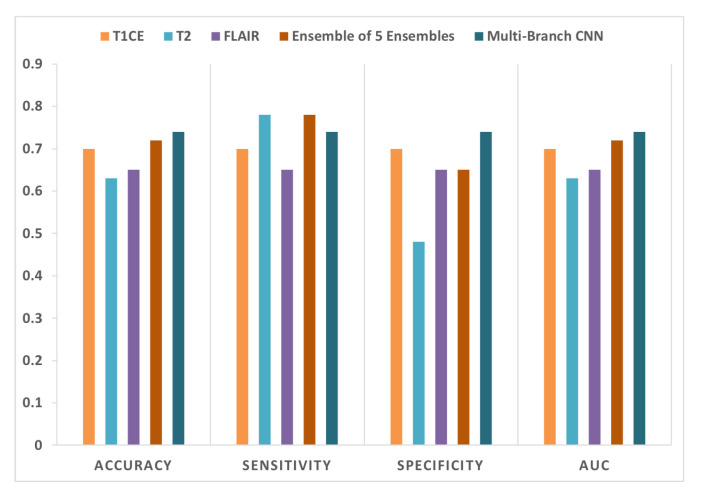
Performance comparison of the multi-branch CNN vs individual sequences vs ensemble of the five voted ensembles method.

**Table 1 diagnostics-12-00345-t001:** Performance results of the ensemble of five snapshots using the 3D T1CE sequence.

Method	Accuracy	Specificity	Sensitivity	AUC
Snapshot 1	63%	39%	87%	0.63
Snapshot 2	67%	74%	61%	0.67
Snapshot 3	70%	74%	65%	0.70
Snapshot 4	61%	74%	48%	0.61
Snapshot 5	59%	52%	65%	0.59
Ensemble	70%	70%	70%	0.70

**Table 2 diagnostics-12-00345-t002:** Performance results of the ensemble of five CNNs (convolutional neural networks) using the 3D T1CE sequence.

Method	Accuracy	Specificity	Sensitivity	AUC
CNN 1	63%	48%	78%	0.63
CNN 2	54%	48%	61%	0.54
CNN 3	61%	78%	43%	0.61
CNN 4	61%	65%	57%	0.61
CNN 5	57%	43%	70%	0.57
Ensemble	67%	70%	65%	0.67

## Data Availability

BraTS dataset can be downloaded online from the CBICA’s Image Processing Portal (IPP) after requesting permission from the BraTS team. Our implementation code is available on GitHub at https://github.com/kbenahmed89/Glioma-SurvivalPrediction, accessed on 14 December 2021.

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
