# Peer review of "Ensembles of Convolutional Neural Networks for Survival Time Estimation of High-Grade Glioma Patients from Multimodal MRI"

_diagnostics, 2022, doi:10.3390/diagnostics12020345_

Round 1
Reviewer 1 Report
I would like to congratulate the authors of the article “Ensembles of Convolutional Neural Networks for Survival Time Estimation of High-grade Glioma Patients from Multimodal MRI” for their exceptional work.
The paper contains a lot of good information. However, some improvements would be necessary.
- Regarding „Figure 1: Different planes of brain MRI” – please specify which MRI sequence is present in these images.
- The application format of the reference section is incorrect and must be revised according to MDPI standards.
- There is a newer edition of the WHO classification of tumors of the central nervous system, published in 2021. Please check it and update reference [1], accordingly.
- In the vast majority of cases, the diagnosis and grading of glioma requires histopathological examination/confirmation. Moreover, there are multiple histopathological and immunohistochemical prognostic factors (Ki67 index, IDH status, etc.) proven to play a significant role in overall patient survival. Although not being the subject of this work, it would be worthwile mentioning their importance in correlation with the imagistic aspects. Recommended reading/reference: Lisievici AC, Lisievici MG, Pasov D, Georgescu TA, Munteanu O, Grigoriu C, Bohiltea RE, Furtunescu FL, Sajin M. Practical aspects regarding the histopathological grading and anaplastic transformation of gangliogliomas – a literature review. Romanian Journal of Morphology and Embryology 2021, 62(2).
- The authors should include suggestion(s) for future direction(s) in the final paragraphs.
Reviewer 2 Report
Introduction: Average survival is now commonly 16 months, not 12 months (quote 1 in the manuscript); you may update this data to the control arm of EF-14 study:
Effect of Tumor-Treating Fields Plus Maintenance Temozolomide vs Maintenance Temozolomide Alone on Survival in Patients With Glioblastoma: A Randomized Clinical Trial. JAMA. 2017 Dec 19;318(23):2306-2316. doi: 10.1001/jama.2017.18718.
Materials and Methods: explain the difference between ET and TC - is tumor core central necrosis?
Results: prediction accuracy is 74% meaning one is couseling 1 out of 4 patients uncorrectly? This is not an acceptable rate in cancer therapy.
Outcome prediction is a relevant information, but mostly for the treating physician and not in many cases for the patients themselves. One never tells a patient: "Based on my calculations I expect you to live 5 months only." So this aspect may be relevant for caregivers and/ or relatives. Only a few patients want to hear an exact figure for survival analysis.
I do have a problem with the general set-up of trying to predict survival on a preoperative dataset. You are leaving out many outcome-relevant factors that develop after histological confirmation and severely influence outcome:
1- clinical data: ECOG score / Karnofsky performance score, age, eloquency of the lesion, postoperative neurological deficit, dexamethasone dose, dose limiting toxicity while under chemotherapy, radiation therapy dose
2- radiological data: multifocality, involvement of subventricular zone, posterior fossa / brain stem involvement, resection status (MRI remnant?), perioperative ischemia in peritumoral areas
3- molecular tumor profile: IDH status, MGMT promotor methylation, histone3 status (H3K27M), methylation class profile
Please discuss in detail how these parameters influence outcome. How about including the postoperative MRI in CNN?
Please speculate on a combined score of clinical data and CNN for survival prediction.
Please speculate on using CNN for histological determintaion instead of biopsy / resection in highly eloquent areas (e.g. brain stem).
What is the role of metabolical imaging (e.g. 2-hydroxyglutarate MR-spectroscopy or amino acid PET) for CNN?
In this manuscript you are shedding light on just one aspect out of more than two handful that influence survival in GBM patients, therefore sensitivity, specificity and AUC are far from precise.
Round 2
Reviewer 2 Report
Accept